# A transportome-scale amiRNA-based screen identifies redundant roles of *Arabidopsis* ABCB6 and ABCB20 in auxin transport

Yuqin Zhang[1], Victoria Nasser[1], Odelia Pisanty[1], Moutasem Omary[1], Nikolai Wulff [2], Martin Di Donato[3], Iris Tal[1], Felix Hauser [4], Pengchao Hao[3], Ohad Roth[1], Hillel Fromm[1], Julian I. Schroeder[4], Markus Geisler[3], Hussam Hassan Nour-Eldin [2] & Eilon Shani[1]

Transport of signaling molecules is of major importance for regulating plant growth, development, and responses to the environment. A prime example is the spatial-distribution of auxin, which is regulated via transporters to govern developmental patterning. A critical limitation in our ability to identify transporters by forward genetic screens is their potential functional redundancy. Here, we overcome part of this functional redundancy via a transportome, multi-targeted forward-genetic screen using artificial-microRNAs (amiRNAs). We generate a library of 3000 plant lines expressing 1777 amiRNAs, designed to target closely homologous genes within subclades of transporter families and identify, genotype and quantitatively phenotype, 80 lines showing reproducible shoot growth phenotypes. Within this population, we discover and characterize a strong redundant role for the unstudied *ABCB6* and *ABCB20* genes in auxin transport and response. The unique multi-targeted lines generated in this study could serve as a genetic resource that is expected to reveal additional transporters.

[1] School of Plant Sciences and Food Security, Tel Aviv University, Tel Aviv 69978, Israel. [2] DynaMo Center, Copenhagen Plant Science Center, Department of Plant and Environmental Sciences, Faculty of Science, University of Copenhagen, Frederiksberg 1871, Denmark. [3] Department of Biology, University of Fribourg, CH-1700 Fribourg, Switzerland. [4] Division of Biological Sciences, Cell and Developmental Biology Section, University of California, San Diego, La Jolla, 92093–0116 CA, USA. Correspondence and requests for materials should be addressed to E.S. (email: eilonsh@post.tau.ac.il)

Plants are complex organisms that have the ability to respond to environmental cues. These responses are mediated by the rapid transport of a wide variety of substrates from one part of the plant to another. In addition to the primary metabolites found in all plant species, there are estimated to be more than 200,000 secondary metabolites[1,2], and these organic molecules regulate all aspects of plant biology. In many cases, plants spatially balance metabolites and restrict them to specific tissues and cell-types to allow proper growth and response to biotic and abiotic stresses[3,4]. The active movements of metabolites fluctuate due to changes in the environment and depending on the developmental stage. A variety of mechanisms are involved in long and short distance transfer of metabolites, and active transporters can regulate both. A few examples are the BOR and NIP boron transporters[5], the NPF nitrogen transporters[6], the GTR glucosinolate transporters[7], the SWEET sucrose transporters[8], and the AAP and LHT amino acid transporters[9]. Active transport of organic small molecules is not restricted to cell-to-cell transport or vasculature loading and unloading but also takes place at the subcellular level to allow intracellular compartment allocation[10–13].

Plant growth and development are mediated to a large extent by a group of small and mobile signaling molecules named hormones. Plants regulate hormone response pathways at multiple levels including biosynthesis, metabolism, perception, and signaling[14–16]. In addition, plants tightly control the spatial hormone distribution[17,18]. This is illustrated most clearly in the case of auxin, where the combined activity of auxin influx and efflux carrier proteins generates auxin maxima and minima that inform developmental patterning. The regulation of the cellular localization of PIN-FORMED (PIN) efflux transporters determines the direction of auxin flow from one cell to another[19,20]. In addition, PILS and ABC family proteins transport auxin[21,22]. More recently, gibberellin (GA), abscisic acid (ABA), strigolactone, and cytokinin transporters from the NPF, ABC, and PUP families have been identified[23–30]. Although multiple studies suggest that individual NPF and ABC genes have specialized functions, genetic analyses of these families have been limited by the scarcity of loss-of-function phenotypes. The major reason for this is that plant genomes contain redundant genetic elements. For example, there are huge numbers of genes encoding NPF and ABC transporters (53 and 147, respectively, in Arabidopsis) that exhibit high redundancy despite diverse substrate specificity of the encoded proteins[31–34].

Over the past 2 decades, forward genetic screens have proven to be powerful approaches in the search for gene function in general and in the isolation of transporters; however, there are no observable phenotypes for many single-gene loss-of-function mutants. As of 2013, only 2400 Arabidopsis genes (~8%) were documented to have a loss-of-function mutant phenotype, and 401 Arabidopsis genes (1.5%) were found to exhibit a mutant phenotype only when disrupted in combination with a redundant paralog[35]. Thus, as evidenced previously discernable phenotypes in plants mutated in single transporters may be masked by functionally redundant gene paralogs[36]. Many gene paralogs are arranged in tandem in plant genomes[35], and functional analysis of tandem paralogs is additionally hampered by the low frequency of recombination between adjacent genes. Altogether, these data suggest that a large fraction of potential phenotypic plasticity is "hidden"[37,38].

Here, we utilize a transportome-scale artificial microRNA (amiRNA) approach[38], to overcome the challenge of functional redundancy in plant transport processes. We generated a unique population of 3000 amiRNA lines wherein each amiRNA was designed to target closely homologous genes within sub-clades in transporter families in a variety of combinations[38]. Our phenotypic screen revealed 95 reproducible shoot phenotypes lines. Among these lines, 80 presented phenotypes that had not been associated with the targeted genes before. Out of the 95 lines, 26 were shown to exhibit differential response to different plant hormone treatments, indicating an involvement in hormone regulation. We chose to characterize one of these lines (amiR1334) due to a striking auxin-related shoot phenotypes. We show that amiR1334 targets the previously unstudied ABCB6 and ABCB20 genes, and through expression analyses and transport assays verify that ABCB6 and ABCB20 are redundantly required for the basipetal movement of auxin in the shoot. By genotyping and phenotyping each of the 95 lines, we have created a genetic resource that will be useful in understanding plant metabolite transport.

## Results

**Transportome multi-targeted amiRNA-based phenotypic screen.** To overcome the potential genetic redundancy among plant transporter families (transportome), we utilized the PHANTOM amiRNA library[38] designed to target Arabidopsis gene families. Hauser et al. generated a genome-scale amiRNA library by synthesizing 22,000 amiRNA constructs that putatively silence 18,117 genes in different combinations with 96% of the amiRNAs co-silencing 2–5 closely related genes[38].

A transporter-specific amiRNAs sublibrary, consisting of 1777 synthetic amiRNAs[38] was transformed in bulk into wild-type (WT) Arabidopsis (Col-0). T$_1$ seeds were selected for BASTA-resistance, and 3000 T$_2$ plants were collected individually. T$_2$ seeds of all 3000 lines were sown in soil and grown under normal conditions. A phenotypic screen was carried out from seedlings stage to flowering, with a focus on selecting phenotypes displaying altered shoot growth, leaf color, and morphology. We expected a dominant phenotype from the amiRNA, but in a few cases (3.6%) we detected clear recessive phenotypes that probably resulted from construct integration into a gene rather than from amiRNA activity. These lines were not included in further work. The screen identified 105 lines with interesting phenotypes. In order to validate the phenotypes, we selected homozygous T$_3$ lines for the 105 lines and re-screened them. Of the 105 lines showing shoot growth or leaf morphology phenotype in the T$_2$ generation, 95 had phenotypes that were reproducible in T$_3$ lines. It is possible that the amiRNAs in the remaining lines were silenced over generations, which is a phenomenon that we often observe. We therefore continued to work only with the reproducible 95 amiRNA lines (Supplementary Table 1). The sequence of the amiRNA was successfully determined in 79 of these unique T$_3$ lines, whereas we could not obtain sequence information of the amiRNA in 16 of the lines. The 79 amiRNAs collectively target 193 genes (Supplementary Table 1). Out of the estimated 1185 known Arabidopsis transporters, this number constitutes approximately 16% of the Arabidopsis transportome. We found 22 genes within 15 lines presenting previously reported phenotypes (Supplementary Figure 1), indicating on the screen strength. For example, we identified line 1162, targeting four PATL genes and line 686, targeting two PATL genes that show shoot growth defects (Supplementary Table 1). This result is in agreement with Tejos et al. who recently showed that while the single patl mutants do not present a phenotype, the quadruple patl2 patl4 patl5 patl6 present reduced PIN1 repolarization in response to auxin and disrupted plant development[39]. We found 43 genes within 33 lines that presented previously reported phenotypes, which were different or partially different from the phenotypes we observed (Supplementary Figure 1). This could result from the co-silencing of additional genes in the family or by characterization of different developmental stages

and environmental conditions. Finally, we found 128 genes within 62 lines (66.3% of the identified genes) presenting previously unknown loss-of-function phenotypes (Supplementary Figure 1). Notably, the 193 target genes should be classified as putative targets as we have not validated off-targets effects for all lines. In addition, the results cannot exclude the possibility that several of the phenotypes are a result of down regulating only one of the target genes. In three different instances we found the same amiRNA transformed into multiple (2 or 3) independent lines showed similar phenotypes (Supplementary Table 1), suggesting that the screen is reliable but not saturated.

To further examine the reproducibility of the PHANTOM transportome amiRNA library results, we re-cloned and transformed plants with seven amiRNAs that showed strong growth phenotypes. Six out of the seven amiRNAs showed reproducible phenotypes that varied between independent transformation lines (Supplementary Figure 3). The observed variation between lines (52–87%) is equivalent to the report of amiRNA functionality and penetration by Schwab et al.[40].

The 193 putative targets genes were classified into 11 groups according to the classification of the transporters database ARAMEMNON[41] (Fig. 1a, Supplementary Table 1). In most groups, the percentage of potentially silenced genes within the family varied between 14 and 21% (Fig. 1b). The porins and p-type ATPase were at the extremes of percent at which phenotypes were observed, but this could not be statistically verified since the number of genes within these groups are relatively low. Interestingly, for the major facilitator superfamily (MFS) and the "other groups" (OG) less than 10% of genes in the family were co-silenced (Fig. 1b). In total, we identified 95 lines with observable phenotypes, among them 80 lines are association of gene function to phenotypes, exposing parts of the plant transportome phenotypic plasticity previously hidden by gene redundancy.

**Phenotypic and hormone response characterization**. Taking advantage of a high-throughput phenomics system, we evaluated photosynthesis, plant color shade, and shoot morphology parameters under normal growth conditions for the 95 homozygous amiRNA lines that showed reproducible growth phenotype. Data are summarized in Supplementary Table 1. Of the 95 lines, 65 showed significant plant height difference compared to WT. Six lines were taller than WT, whereas 59 lines were shorter (Fig. 1c). Color shade analysis showed several lines with differences in leaf color compared with WT plants. The color of amiRNA-32 leaves is visibly darker, whereas the color of amiRNA-426 leaves is visibly lighter (Fig. 1d, Supplementary Figure 2). In agreement, amiRNA-426 targets two genes, among them the ABC transporter ATM3, of which its mutant presents dwarf and chlorotic plants[42]. Thirty-two amiRNA lines showed significant differences in shoot size compared to WT (Fig. 1e). Among these lines, 30 had smaller and two had larger shoots than WT. To better characterize the phenotypes of the amiRNA lines, other morphological parameters were analyzed, including rosette perimeter, eccentricity, rotational mass symmetry, index of rosette shape, and compactness (Supplementary Figure 4).

In addition, we analyzed a number of photosynthesis parameters including fluorescence decline ratio in steady-state (Rfd_Lss). Among the 22 lines showing significant differences compared to WT, four lines showed lower Rfd_Lss activity, which might indicate for smaller stomatal conductance, and 18 lines had higher Rfd_Lss, that might be linked to larger stomatal conductance under normal conditions (Fig. 1f). Additional photosynthesis parameters such as maximum PSII quantum yield were also evaluated (Supplementary Figure 5). Due to the

wide range of phenotypes, these lines are a unique genetic resource that can be utilized by the community to further understand plant transport processes.

We were specifically interested in identifying new plant hormone transporters. We hypothesized that amiRNA lines showing resistance or hypersensitivity to specific plant hormone treatments might indicate direct or indirect regulation of a particular plant hormone transport. We therefore screened all 95 amiRNA lines by measuring root growth on auxin (IAA) gibberellin (GA$_3$), abscisic acid (ABA), cytokinin (t-Zeatin), and jasmonic acid (JA). We identified 32 amiRNA lines with partial hypersensitivity or insensitivity to different phytohormone treatments. In order to validate these hormone-related sub-screen results, we repeated all assays for all 32 lines. Twenty-six lines showed reproducible hypersensitivity or insensitivity to different phytohormone treatments; IAA: 7 lines; GA$_3$: 3 lines; ABA: 7 lines; t-Zeatin (CK): 6 lines; and JA: 7 lines (Fig. 2, Supplementary Figure 6). It should be noted that these results do not necessarily indicate that the amiRNA-targeted gene has a direct role in hormone transport. However, this sub-screen served as a starting point for identification of putative hormone transporter groups.

**ABCB6 and ABCB20 redundantly regulate shoot growth**. amiR1334 (amiRNA-1334 line) presented a striking developmental phenotype of shoot growth inhibition and small leaves (Fig. 3a, Supplementary Figure 7A). As the plants progressed through developmental stages, the amiR1334 phenotype was enhanced showing short inflorescence stems and skewed small leaves (Fig. 3a, Supplementary Figure 7A, Supplementary Figure 8D, F), similar to known defects in auxin responses and distribution[43,44]. Interestingly, amiR1334 targets two closely related genes from the ABCB family: ABCB6 (AT2G39480) and ABCB20 (AT3G55320) (Fig. 3b). These genes are of particular interest since several members of the ABCB family, including ABCB1, ABCB19, ABCB21, and ABCB4 have been shown to mediate auxin transport through the plasma membrane[45–53]. The functions of the closely related ABCB6 and ABCB20 have not been previously studied nor linked to auxin transport. The amiR1334 target sequence has two nucleotide mismatches with ABCB6 and ABCB20, and five mismatches with ABCB1 (AT2G36910), including a mismatch in the 3′ region of the seed sequence, and no additional targets are detected in silico (Supplementary Figure 7b). Therefore, we speculated that the phenotype of amiR1334 is predominately caused by the down-regulation of ABCB6 and ABCB20.

In order to determine if the amiR1334′s target genes (ABCB6 and ABCB20) are involved in auxin response, we generated homozygous T-DNA-insertion lines for ABCB6 and ABCB20 (Supplementary Figure 8a) and performed a hypocotyl elongation assay on the amiR1334 line, the single T-DNA mutant lines and WT (Fig. 3c). Plants were transferred to Murashige-Skoog plates with 5 μM IAA, 3 μM NAA, or 100 nM of the auxin derivative 2,4-dichlorophenoxyacetic acid (2,4-D). Hypocotyls of amiR1334 seedlings showed a normal response to IAA, but a reduced response to the synthetic auxin NAA and 2,4-D treatment (Fig. 3d). A similar trend was observed in the single T-DNA-insertion mutants of ABCB6 and ABCB20 (Fig. 3d). In addition, the amiR1334 line showed shoot growth hypersensitivity to the auxin transport inhibitor N-1-naphthylphthalamic acid (NPA) (enhanced shoot growth inhibition) (Supplementary Figure 8b,c). The partial insensitivity of amiR1334 to synthetic auxin and the hypersensitivity to NPA suggested that ABCB6 and ABCB20 may be involved in auxin response in the shoot.

In order to genetically verify the cause of the amiR1334 phenotype and assess the specific contribution of ABCB6 and

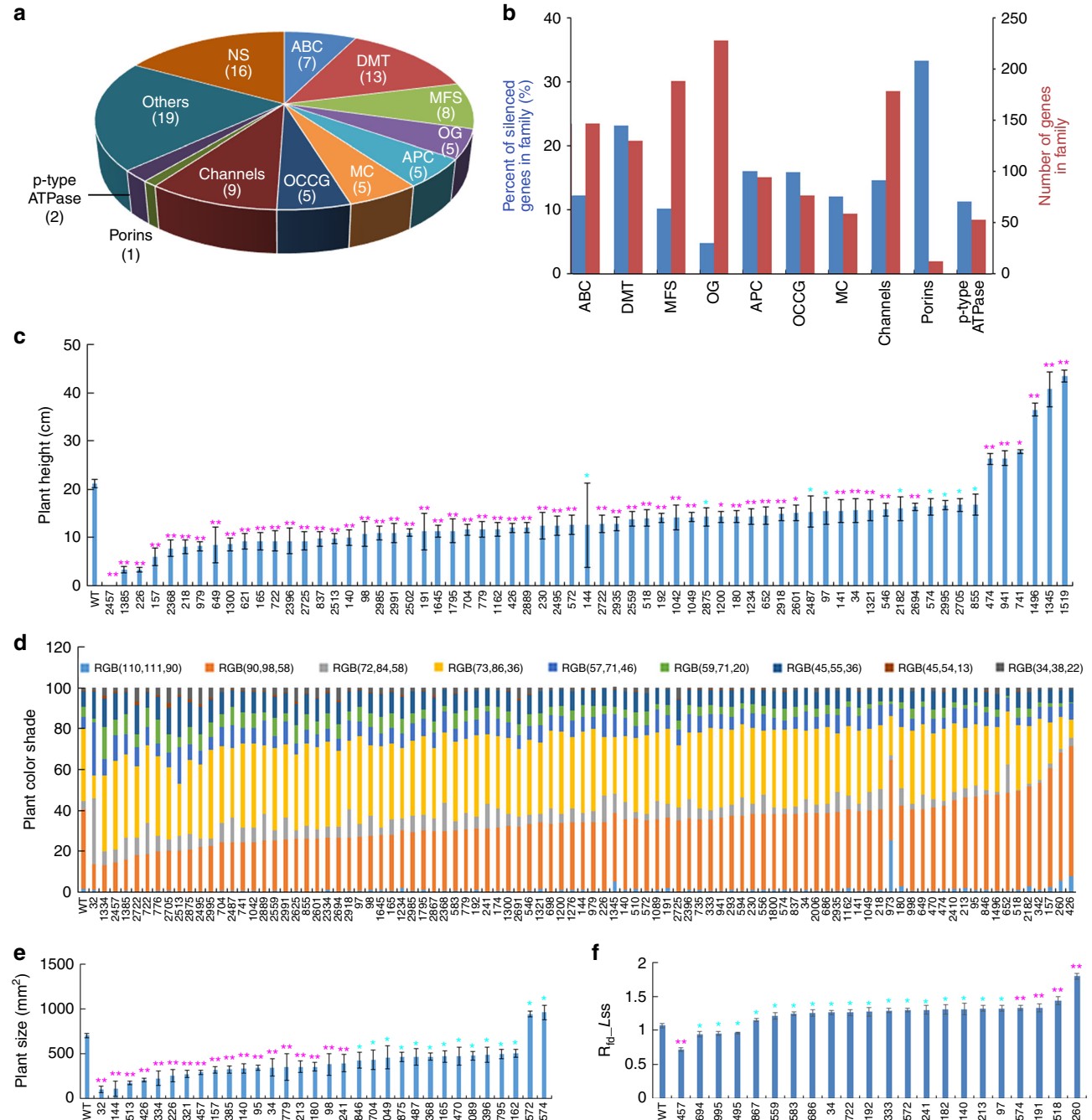

**Fig. 1** Multiplexed transportome-specific amiRNA screen reveals novel phenotypes of redundant genes. **a** The number of different groups of amiRNA lines with significant phenotypes obtained from the transporter library. *Abbreviations*: ABC ATP-binding cassette transporter, DMT drug/metabolite transporter, MFS major facilitator superfamily, OG other groups, APC amino acid/polyamine/organo-cation, OCCG other cation carrier groups (i.e., CPA, CaCA, CDF, CaCA2), MC mitochondrial carrier, Others (additional transporters that do not belong to families mentioned above), NS non-sequenced. **b** The percent of amiRNA target genes within the transporter family (blue bars), and the number of transporters in each transporter family (red bars). **c** Heights of 50-day-old amiRNA lines presenting significant phenotypes ordered by height. **d** Color shade of all 25-day-old amiRNA lines ordered by RGB score. **e** Plant shoot size (rosette leaves area) of 25-day-old amiRNA lines showing significant phenotypes. **f** Fluorescence decline ratio in steady state (Rfd_Lss) of 25-day-old amiRNA lines showing significant phenotypes. For all panels, shown are averages (±SE) of 5 plants, differences are significant at *$P < 0.05$ and **$P < 0.01$ by Student's *t*-test

*ABCB20* genes to the phenotype, we generated single and double mutant T-DNA *abcb6-1 abcb20-1* lines (Fig. 3c). Whereas the single *abcb6-1* (GABI_5401D12) and *abcb20-1* (GABI_520G10) mutants did not show any significant growth phenotype, the *abcb6-1 abcb20-1* double mutant exhibited a strong developmental phenotype that mimicked that of the *amiR1334* line with dwarf stature and small skewed leaves (Fig. 3e, f, Supplementary

Figure 8D,F). We further obtained and genotyped an additional, independent *abcb20* allele (GABI_387F09) termed *abcb20-2* (Fig. 3c, Supplementary Figure 9A). The *abcb6-1 abcb20-2* double mutant showed identical phenotypes to *abcb6-1 abcb20-1* double mutant and *amiR1334* line (Fig. 3g, Supplementary Figure 8D, F, Supplementary Figure 9B-E). Nanostring experiments showed that transcript levels of *ABCB6* and *ABCB20* are significantly

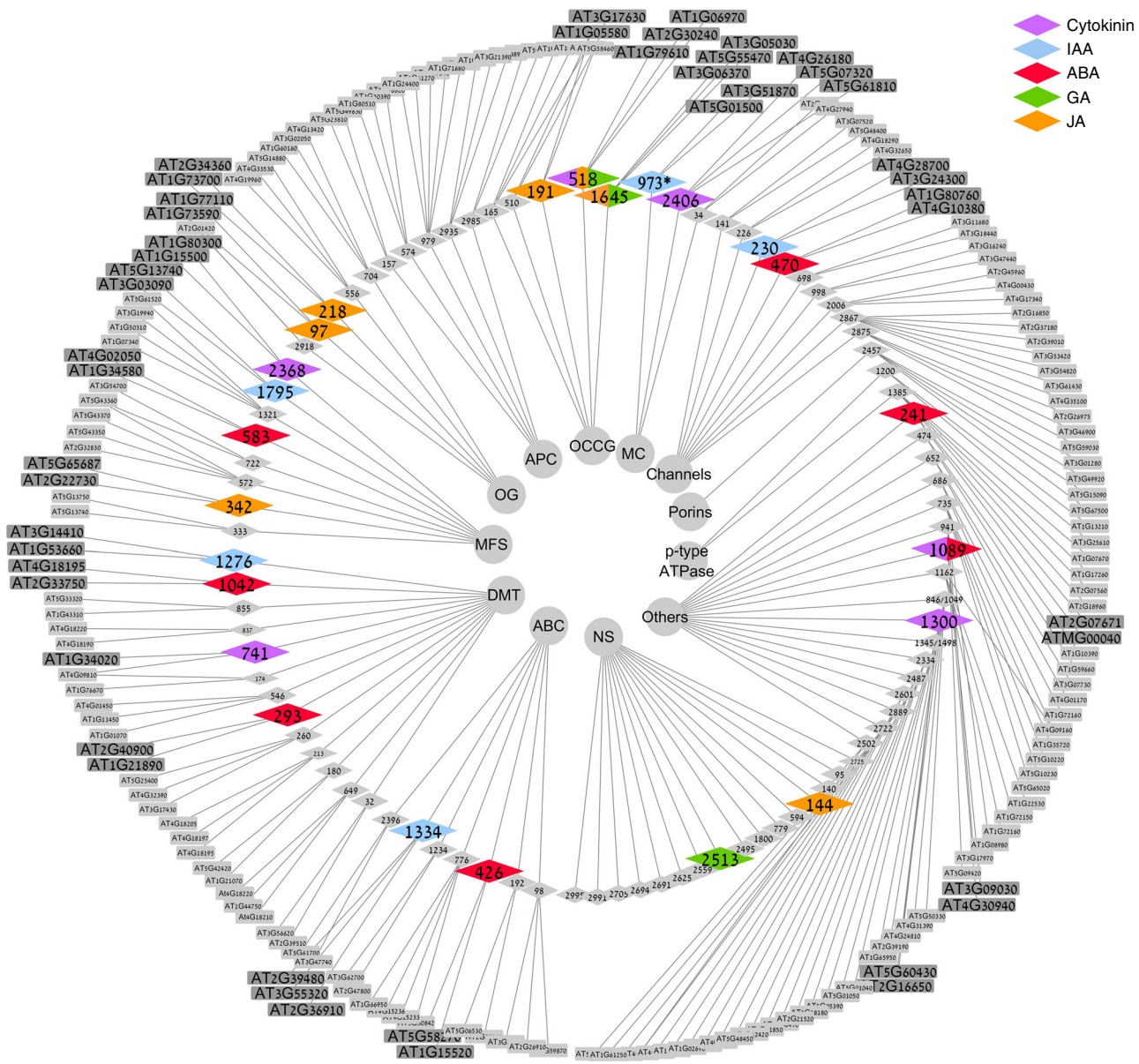

**Fig. 2** Transporter-targeted amiRNA lines show significant developmental responses to plant hormones. Gray lines connect transporters groups, amiRNA lines, and their putative gene targets. Transporter groups are indicated by the inner gray circles. amiRNA lines showing significant developmental responses to particular hormones are indicated by enlarged color-coded diamonds (middle circle). Putative amiRNA target genes are indicated in enlarged gray squares (outer circle). amiRNA lines with insignificant developmental responses to hormones are indicated by the small gray diamonds (middle circle) with putative amiRNA target genes indicated by small gray squares (outer circle). amiRNA designated 973* indicates for three independent amiRNA lines (973, 2182, 2410) with the same target sequence in Supplementary Table 1. *Abbreviations*: ABC ATP-binding cassette transporter, DMT drug/metabolite transporter, MFS major facilitator superfamily, OG other groups, APC amino acid/polyamine/organo-cation, OCCG other cation carrier groups (i.e., CPA, CaCA, CDF, CaCA2), MC mitochondrial carrier, Others (additional transporters that do not belong to families mentioned above), NS non-sequenced

down regulated in both alleles of *abcb6 abcb20* double mutant and *amiR1334* line (Fig. 3h, Supplementary Figure 9F, Supplementary Figure 13B). The transcript levels of the related auxin transporters *ABCB1* and *ABCB19* did not change in these lines (Supplementary Figure 10a,b). This result suggests that the phenotype of the *amiR1334* line is due to down-regulation of *ABCB6* and *ABCB20* transcripts and that ABCB6 and ABCB20 share overlapping activities.

We further tested the expression level of all ABCBs in *amiR1334* plants and *abcb6-1 abcb20-1* compared to WT. While *ABCB6* and *ABCB20* were significantly down regulated in both *amiR1334* and *abcb6-1 abcb20-1* compared to WT, four

additional *ABCBs* (*ABCBs 5,7,16,17*) were slightly down regulated (non-significant) and two *ABCBs* (*ABCB18* and *22*) were slightly up-regulated (non-significant) (Supplementary Figure 11).

Auxin regulates many aspects of plant growth and development including phyllotaxis[54]. Interestingly, the *abcb6-1 abcb20-1* double mutant showed strong phyllotaxis defects as siliques initiated in a non-spiral phyllotactic arrangement with an average angle of 150° compared to 134° in the WT (Fig. 3i, j). The defects in phyllotaxis were observed both in angle and in spacing between lateral organ initiation sites (Supplementary Figure 12a). However, the defects on angle organ initiation is also attributed by the twisting of the stem in *abcb6-1 abcb20-1* double mutant

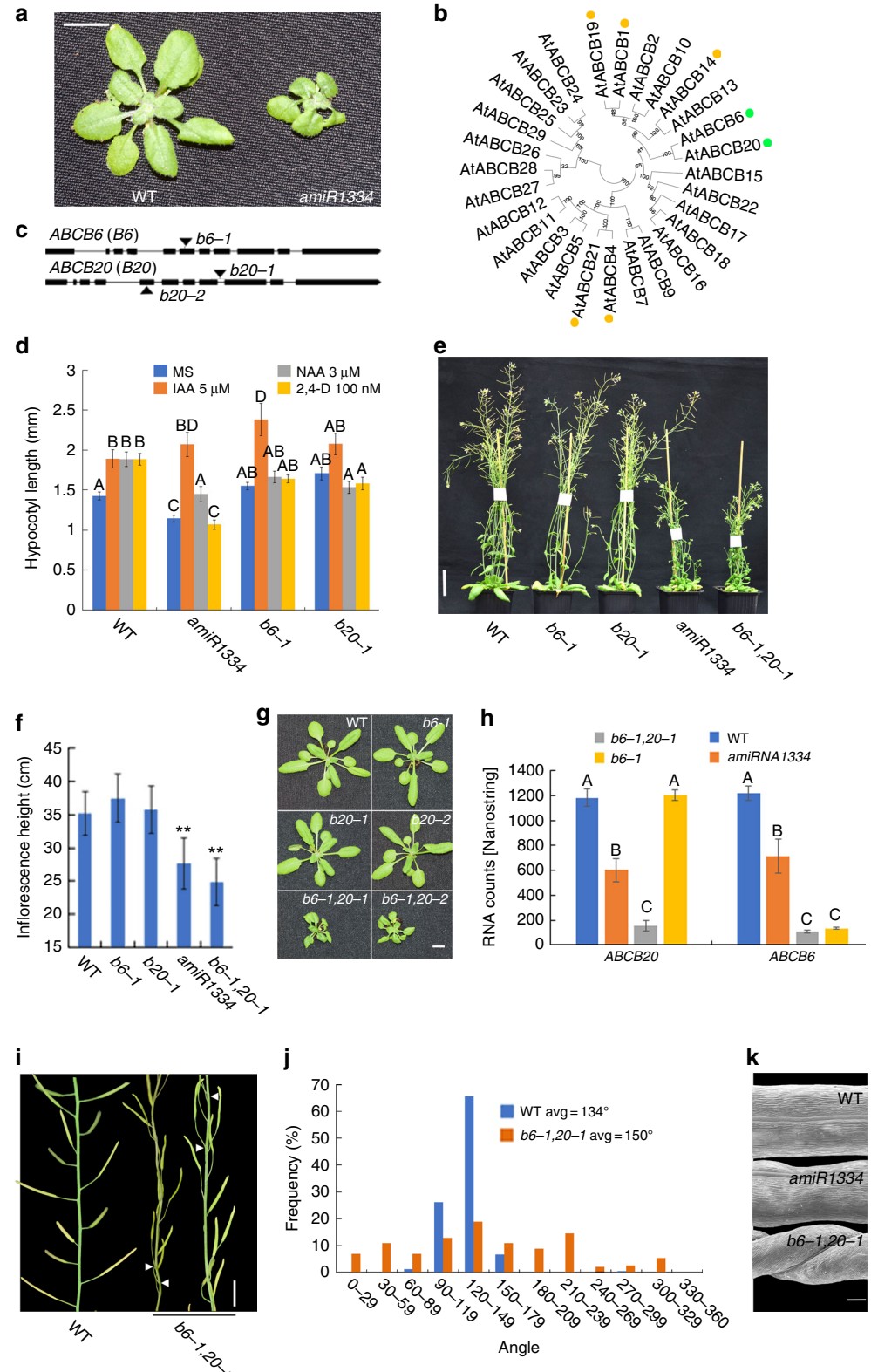

(Supplementary Figure 12b,c). In addition, the siliques of *amiR1334* plants were slightly twisted and the siliques of *abcb6-1 abcb20-1* double mutant plants were highly twisted (Fig. 3k), possibly due to uneven growth of the epidermis and inner mesophyll layers as reported for *abcb1 abcb19* and *twd1* mutants[55]. The *amiR1334* and *abcb6 abcb20* double mutant but not the single mutants showed weak but significant root growth

phenotype (Supplementary Figure 8e). In summary, these findings strongly suggest that ABCB6 and ABCB20 redundantly function to regulate shoot growth.

**ABCB6 and ABCB20 function redundantly in auxin export.** In order to determine if ABCB6 and ABCB20 facilitate auxin

**Fig. 3** ABCB6 and ABCB20 redundantly regulate *Arabidopsis* shoot growth. **a** Shoot phenotypes of 25-day-old WT and *amiR1334* seedlings grown on soil under normal conditions. Scale bar, 1 cm. **b** Phylogenetic tree of *Arabidopsis* ABCBs. Orange dots indicate previously characterized auxin transporters. Green dots indicate putative transporters characterized in this study. **c** Representation of the *ABCB6* and *ABCB20* genes. Arrows indicate the position of T-DNA insertion. **d** Hypocotyl growth of the indicated genotypes. Seedlings were grown on 1/2 MS for 4 days and transferred to the indicated treatments. Hypocotyls were imaged and lengths quantified at day 7. Shown are averages (±SE), letters indicate statistically significant differences at $P < 0.05$. $n = 20$ plants. **e** Phenotypes of mature 6-week-old plants grown in soil. Scale bar, 5 cm. **f** Quantification of inflorescence height of the genotypes shown in (**e**). Shown are averages (±SE), $n = 16$ plants, **$P < 0.01$, Student's *t*-test. **g** Phenotypes of 35-day-old plants grown in soil. Scale bar, 1 cm. **h** *ABCB6* and *ABCB20* expression level for the indicated genotypes (12-day-old seedlings). $n = 3$ biological replicates. **i** Phyllotaxis of the *b6-1,20-1* double mutant (two plants at the right) compared to WT (left). Scale bar, 1 cm. **j** Quantification of phyllotaxis shown in (**i**), presented as angle frequencies. $n = 256$ for WT and 249 for *b6-1,20-1* plants. **k** Silique morphology of the indicated genotypes. SEM images, scale bar, 200 μm. *Abbreviations*: *b6* and *b20* stands for *abcb6* and *abcb20* single mutants, respectively. *b6-1,20-1* stands for *abcb6-1 abcb20-1* double mutant. *b6-1,20-2* stands for *abcb6-1 abcb20-2* double mutant

transport, as suggested by loss-of-function phenotypes and activities of the characterized family members, we carried out auxin transport and response assays in whole plants and protoplasts. First, we utilized the auxin reporter *pDR5:GUS* and evaluated the response to auxin following IAA treatment on the apical side of the inflorescence stem (Fig. 4a, b). Strong GUS staining was detected after 5.5 h at the basal side in WT background, but in contrast, no signal was detected from *pDR5:GUS* in the *amiR1334* background (Fig. 4b, Supplementary Figure 13a), suggesting that basipetal auxin transport was disrupted in this line. Next, we analyzed polar auxin transport in stems by applying radiolabeled IAA to the apical or basal side of the inflorescence stem and detected its accumulation on the opposing side after 4 h. This showed a statistically significant reduction (~50%) in polar auxin transport in the inflorescence stems of the double mutant *abcb6-1 abcb20-1* compared to that in WT (Fig. 4c), in agreement with a significant *ABCB6* and *ABCB20* transcriptional downregulation in stem segments compared to WT (Supplementary Figure 13a). This indicates that ABCB6 and ABCB20 are essential for basipetal polar auxin transport. We furthermore tested export of radiolabeled IAA in a cellular system using isolated *Arabidopsis* leaf mesophyll protoplasts[48]. In this system, *abcb6-1* and *abcb20-1* single mutants showed a 54% and 70% reduction in export activity, respectively, compared to WT, while the *abcb6-1 abcb20-1* double mutant and the *amiR1334* auxin export was lower by about 73% and 66% compared to that in WT protoplasts (Fig. 4d). We further tested the export of benzoic acid (BA), which is often used as a diffusion control in auxin research and malic acid, which was shown to be transported by ABCB14[56]. Transport of both substrates was not significantly different in *abcb6-1* and *abcb20-1* mutant protoplasts indicating the specificity of ABCB6 and ABCB20-mediated auxin transport (Supplementary Figure 14a,b). Altogether, these results suggest that ABCB6 and ABCB20 specifically promote auxin export and act redundantly to regulate shoot growth.

In order to better understand how ABCB6 and ABCB20 participate in the cellular regulation of auxin transport activity we tested their subcellular protein localization. We were unable to amplify and clone the *ABCB20* coding or genomic sequences. However, the *ABCB6* coding sequence was fused to *YFP* and cloned under the control of a *35S* promoter (*p35S:ABCB6-YFP*). The vector was then transiently expressed in *Arabidopsis* protoplasts and also via infiltrated tobacco leaves. ABCB6 showed plasma membrane localization both in *Arabidopsis* (Fig. 4e) and tobacco (Fig. 4f) as it co-localized with FM4-64 in *Arabidopsis* protoplasts and AHA2-RFP plasma membrane marker[57] in tobacco leaves.

To further examine *ABCB6* and *ABCB20* expression patterns we used promoter fragments to drive *GUS-GFP* reporters. Transgenic plants expressing *pABCB6:GUS-GFP* showed a GUS signal in true leaves and the differentiated root vasculature tissues (Fig. 4g). Consistent results of *ABCB6* expression in root

vasculature were obtained by imaging *pABCB6:YFP* plants (Fig. 4h). *pABCB20:GUS-GFP* showed a similar GUS signal in leaves (Fig. 4i). In addition, auxin treatment did not affect the expression levels of *ABCB6* and *ABCB20* (Supplementary Figure 15). Altogether, these results indicate that ABCB6 is plasma membrane localized and that, together with ABCB20, is required for auxin transport to regulate shoot growth and development.

## Discussion
As sessile organisms, plants are particularly susceptible to the ever-changing environment and other external conditions. In order to compete genetically under such conditions, plants have developed robust genetic functional redundancy at the expense of gene specialization[37]. Functional redundancy can be achieved by two mechanisms: First, multiple parallel metabolic pathways that eventually lead to the same outcome can be driven by different genes. Second, multiple homologous genes, resulting from gene duplication, can perform the same or overlapping functions. Retaining duplicated genes can be beneficial as these duplicates buffer fundamental processes from the detrimental effects of random mutations[58]. Plant genomes are highly redundant as over 75% of *Arabidopsis* genes belong to families with at least two members[38]. As a result, most single null mutants do not present an evident phenotype as the overlapping function of one (or more) paralogs might mask phenotypic effects. This fact limits the utility of classic forward genetic studies, which use random mutagenized lines created by chemical or radiation treatments, to identify the players responsible for different processes in plant development and response to the environment.

In order to reveal the functions of redundant genes, Hauser et al. developed and produced libraries of amiRNAs targeting different gene families[38]. By utilizing Hauser's amiRNA plasmid library targeting genes of the transportome, we identified 80 amiRNA lines presenting novel phenotypes. In this screen, the sequence of 79 amiRNA collectively targeted 193 genes which constitutes approximately 16% of the known *Arabidopsis* transporters (estimated 1185 transporters in *Arabidopsis*). Classified into 11 groups according to the classification on ARAMEMNON[41] (Fig. 1a), the percentage of potentially silenced genes within the family varied between estimated 14 and 21% in most groups (Fig. 1b). This suggests that genetic redundancy plays an important role across the transportome. The porins and p-type ATPase were at the extremes of percent at which phenotypes were observed, but this could not be statistically verified since the number of genes within these groups are relatively low. Interestingly, for the MFS and the OG less than 10% of genes in the family were co-silenced (Fig. 1b). One explanation for this difference is that these families have relatively low redundancy, in other words high specification. Another explanation is that these

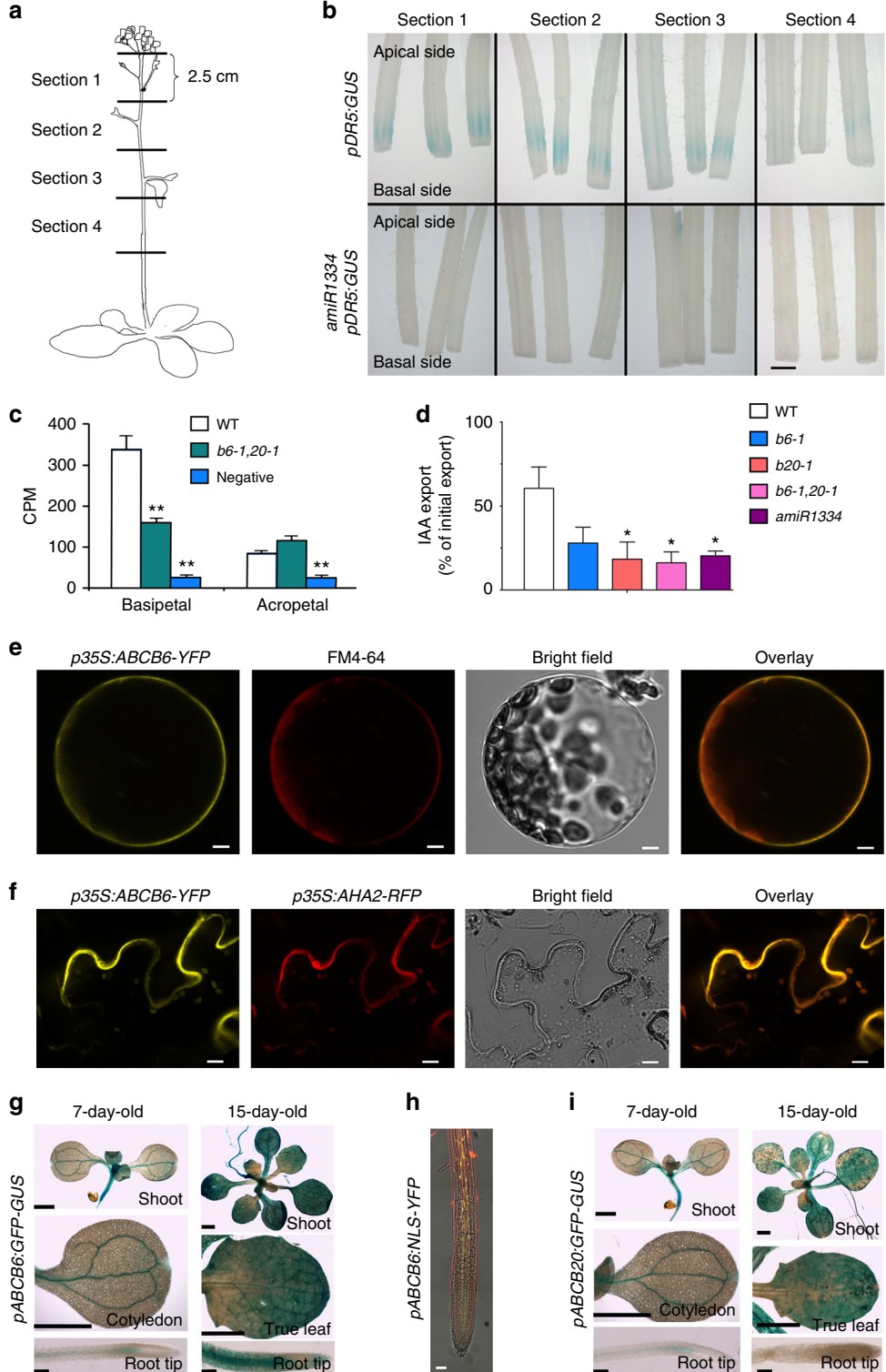

families are too redundant to be exposed in our experimental design.

The evolutionary conserved ABC family is one of the largest transporter families in plants, and plant ABCs have been shown to transport diverse hormones. For example, ABCB isoforms have been demonstrated to transport auxin, while some ABCG isoforms transport cytokinin and ABA as well as other compounds (reviewed in ref. [59]). The redundant auxin transport activity of ABCB1 and ABCB19 and redundant arsenic tolerance of ABCC1

and ABCC2 in Arabidopsis are an example of phenotypic robustness of this family[50,60]. Therefore, the ABCs are an extreme example of gene expansion that provides redundantly robust responses and diverse specialization in plant hormone transport.

Unlike single T-DNA mutants of ABCB6 and ABCB20, the abcb6 abcb20 double mutant showed a strong developmental phenotype revealing a very small stature, curly, skewed leaves, and shorter inflorescence stems, which resembled that of amiR1334 plants suggestive of defects in auxin transport. This

**Fig. 4** ABCB6 localizes to the plasma membrane and promotes auxin transport together with ABCB20. **a** Schematic diagram of plant sections used in inflorescence stem polar auxin transport assays. Stem segments of 2.5 cm were dissected as illustrated and used in the auxin reporter *pDR5:GUS* experiments and radioactive IAA experiments (**b**, **c**). **b** *pDR5:GUS* reporter activity was imaged in response to IAA application to the apical side of the segment. *amiR1334* shows reduction of *pDR5:GUS* activity at the basal side. Section numbers correspond to the section positions shown in (**a**). **c** Basipetal and acropetal radiolabeled IAA ([$^{14}$C]IAA) transport in *Arabidopsis* inflorescence stems. Apical 2.5-cm stem segments were treated with [$^{14}$C]IAA for 4 h. Negative indicates samples not treated with [$^{14}$C]IAA. CPM stands for counts per minute. $n = 10$–15, $**P < 0.01$, Student's *t*-test. **d** [$^{3}$H]IAA export from *Arabidopsis* leaf mesophyll protoplasts. $n = 4$, $*P < 0.05$, Welch's *t*-test. **e**, **f** ABCB6 plasma membrane localization shown by *p35S:ABCB6-YFP* in *Arabidopsis* protoplast (**e**) and tobacco leaves (**f**). Yellow indicates for *ABCB6-YFP* fluorescence, red indicates for FM4-64 (**e**) or the plasma membrane marker AHA2-RFP (**f**). Scale bar, 5 μm (**e**) and 10 μm (**f**). **g** Expression pattern of *pABCB6:GFP-GUS*. Presented are 7- and 15-day-old seedlings. Scale bar, 1 mm for shoot and cotyledon, 0.1 mm for root tip. **h** *pABCB6:NLS-YFP* plants presenting *NLS-YFP* signal (yellow) in the root vascular tissues. Propidium iodide in red. Scale bar, 10 μm. **i** Expression pattern of *pABCB20:GFP-GUS*. Presented are 7- and 15-day-old seedlings. Scale bar, 1 mm for shoot and cotyledon, 0.1 mm for root tip

result strongly indicates that the homologous genes, *ABCB6* and *ABCB20*, are functionally redundant and have a crucial role in shoot growth.

Functional redundancy of ABCB6 and ABCB20 in auxin transport was further supported by quantification of developmental phenotypes of *amiR1334* and of the single T-DNA mutants on various natural and synthetic auxins as well as on auxin transport inhibitors. The shoot surface area of *amiR1334* and the single T-DNA mutants grown on the non-competitive auxin efflux inhibitor, NPA, was significantly smaller than with mock treatment indicating hypersensitivity to NPA (Supplementary Figure 8b,c). Since NPA inhibits ABCB protein function in auxin transport[36], the combination of down-regulating *ABCB6* and *ABCB20* activity together with NPA treatment (which has multiple ABCB targets[61]), resulted in a synergistic effect and a hypersensitivity of *amiR1334*. In another classical auxin-related assay, hypocotyl elongation of *amiR1334* and the single *abcb6* and *abcb20* mutants did respond to IAA but not to the synthetic auxins, 2,4-D and 1-NAA (Fig. 3d), both shown to either not act as an ABCB substrate or to bypass carrier-mediated import by simple diffusion[48]. In most of the phenotypic and transport assays, the impact in *abcb6 abcb20* is similar to *amiR1334* (Fig. 3, Supplementary Figure 7-9), suggesting that the phenotype generated by *amiR1334* is indeed caused by silencing of its putative targets ABCB6 and ABCB20. Although the results show some specificity towards auxin transport activity (ABCB6 and ABCB20 did not affect the transport of BA and malic acid), it is possible that these ABCBs may have additional substrates; dual or multiple substrate specificities are not uncommon among ABCB transporting proteins[22]. It is unclear at this point whether ABCB6 and ABCB20 have further redundant activities with other known ABCB auxin exporters, such as ABCB1 and ABCB19.

The amiRNA screen offers many advantages over classical forward genetic screens. Unlike chemical mutagenesis, often utilized in classical forward genetic screens, amiRNAs are gene-specific and a single amiRNA can be designed to target multiple, related genes. The ease of identification of silenced genes in the amiRNA lines is another huge advantage. In addition, there are multiple examples of gene duplication that result in tandem duplicates in the genome; often these genes are highly redundant. Since these genes are strongly linked, it is very complicated to create double and triple mutant T-DNA-insertion lines. The amiRNA approach overcomes this limitation of targeting multiple linked genes while enabling forward genetic screens.

The amiRNA library screen is a new tool that needs to be further studied, however, we do not know, for example, the off-target rate[40] for the different amiRNAs investigated here, which will lead to noise in the screen. Off-target effects have been extensively explored in the CRISPR system for genome editing[62] and should be evaluated in amiRNA systems as well. In addition,

the fact that amiRNAs can target multiple homologous genes at once is a two-sided coin. The amiRNA library used in the present screen was designed such that 96% of the amiRNAs are predicted to target 2–5 genes[38]. This makes it an outstanding tool to reveal functions of unknown redundant genes. amiRNAs produce knock-down mutants, and are predicted to minimize generation of lethal knock-out mutations[38,40]. Nevertheless, lethality cannot be completely excluded. This could be partially overcome by expressing amiRNAs under an inducible promoter. In addition, from our observations, activities of a few amiRNA lines weakened over generations. This might be a result of silencing of amiRNA expression. To overcome this difficulty for a specific line one can reproduce the phenotype with additional methods that result in stable mutations such as T-DNA and CRISPR. To overcome the silencing issue at the screening level the amiRNA could be expressed under an inducible promoter.

In some rare cases, the amiRNA insert produces a loss-of-function mutant where the phenotype results from interference with the coding sequence in the genome thus knocking out a gene rather than down regulating expression of other genes through an RNA interference mechanism. We observed 3.6% recessive mutations in our screen. These are easily identifiable, since the mutation, in most cases, is recessive and the phenotype is present in only 25% of T$_2$ plants (in contrast to the dominant amiRNA effect which appears in 75% of T$_2$ plants). By re-cloning and transforming several amiRNAs into plants we showed that the transportome amiRNA screen results are highly reproducible (6 out of 7 lines). However, among these lines, we observed significant variation between independent transformation lines, as 52–87% of the lines showed the expected phenotype. While such variation was previously reported for amiRNA activity in plant[40], one might take this into account in future work and should examine multiple independent transformation lines (around 50 lines).

Finally, we found that the most challenging step in the screen was designing the screen in a way that allows the isolation of lines directly involved in the desired process. For example, it is likely that only a few of the 26 lines identified here that show significant hormone responses are indeed the direct down regulation of hormone transporters. Image-based screens using specific fluorescent markers or selection-based screens might be a powerful tool to combine with the amiRNA multi-targeted approach.

In summary, we showed that ABCB6 and ABCB20 redundantly function to regulate shoot growth and likely function as auxin transporters involved in basipetal auxin transport. We expect that the resources generated here linking uncharacterized genes with novel quantitative phenotypes, arising from multi-targeted gene silencing, will be utilized by the community to reveal new mechanisms in hormone signaling and small molecule transport.

## Methods

**Plant material and growth conditions**. All *Arabidopsis thaliana* lines used in this work are in Columbia background (Col-0 ecotype from Salk Institute). For assays on plates, sterilized seeds were plated on $16 \times 16$ cm square petri dishes with growth media containing $0.5 \times$ Murashige-Skoog (MS) medium (pH 5.7, 1% sucrose, 0.8% plant agar). The seeds were stratified for 2–3 days at 4 °C then transferred to growth chambers (Percival, CU41L5) at 21 °C, 100 µE m$^{-2}$ S$^{-1}$ light intensity under long-day conditions (16 h light/8 h dark). For seed production, transformation, crossing, and soil pot assays, seeds were sown onto wet soil, and plants were grown in growth rooms under long day conditions (16 h light/8 h dark) at 21 °C. Sequence data for *Arabidopsis* genes used in this study can be found in the Arabidopsis Genome Initiative under the following accession numbers: *ABCB6* (AT2G39480) and *ABCB20* (AT3G55320).

**amiRNA transportome library generation**. A plasmid library expressing a pool of amiRNAs was designed and synthesized as described in Hauser et al.[38]. *Agrobacterium* (GV3101) was transformed with Plasmid pSOUP. The amiRNA library, constructed in pGREEN was then transformed into the *Agrobacterium* and grown on antibiotic selections (Gentamycin, Rifampicin, Spectinomycin, and Tetra-cycline). The *Agrobacterium* plasmid library was transformed in bulk into six trays of *Arabidopsis* Col-0 plants. Seeds were collected in bulk and sown on 120 trays with soil and sprayed with BASTA for selection at the age of 2 weeks. About 3000 resistant transgenic plants were propagated, and T$_2$ seeds were collected from individual lines. Phenotypic screens were done on the T$_2$ generation. Lines showing a reproducible phenotype were selected, and phenotypes were characterized in T$_3$ homozygous plants.

**Agrobacterium transformation**. Electro-competent *Agrobacterium tumefaciens* strain GV3101 was incubated on ice with 100 ng plasmid for 5 min, then electroporated in a MicroPulser (Bio-Rad Laboratories) (2.2 kV, 5.9 ms). Bacteria were transferred immediately to 1-ml liquid LB tubes and shaken for 2 h at 28 °C. Bacteria were then plated on selective LB agar plates containing the relevant antibiotics for 2 days at 28 °C.

**Arabidopsis transformation**. An *Agrobacterium* colony was chosen and verified by colony PCR and sequencing before growing in 150 ml LB medium containing 25 µg/ml gentamycin and 50 µg/ml rifampicin + construct specific antibiotic for 2 days at 28 °C. Samples were centrifuged for 15 min at 4000 rpm, supernatant was discarded, and the pellet was resuspended in 60 ml 5% sucrose + 0.05% Silwet L-77. *Arabidopsis* flowers were then sprayed with the bacterial solution. After spraying, plants were kept in the dark overnight and grown until siliques ripened and dried. T$_1$ seeds were collected in bulk and sown on MS media containing the appropriate antibiotics for transformant plant selection. Resistant plants were transferred to soil and grown until maturity for propagation.

**Genotyping**. amiRNA lines were genotyped by PCR using primers listed in Supplementary Table 2. The sequence obtained was aligned with the miRNA319 backbone and the 21 amiRNA base pairs were used to identify putative gene targets using the PHANTOM database[38]. T-DNA lines for the single mutants, listed in Supplementary Table 3, were ordered from Gabi Kat (https://www.gabi-kat.de) and The Arabidopsis Information Resource (https://www.arabidopsis.org/). Primers for the T-DNA genotyping were designed using the T-DNA Primer Design Tool powered by Genome Express Browser Server (GEBD) (http://signal.salk.edu/tdnaprimers.2.html). Homozygous mutants were selected by PCR performed with primers listed in Supplementary Table 4.

**Cloning**. *ABCB6* CDS was cloned from Col-0 cDNA using primers listed in Supplementary Table 5. *ABCB6* and *ABCB20* promoters (2323 bp upstream to ATG start codon of *ABCB6* and 638 bp for *ABCB20*) were cloned using primers listed in Supplementary Table 5. Phusion high-fidelity polymerase (New England Biolabs) was used and fragments were then cloned into pENTR/D-TOPO (Invitrogen K2400) and subsequently cloned into binary destination vectors using LR Gateway reaction (Invitrogen 11791) and verified by sequencing.

**Phenomics**. Morphological and photosynthesis parameters were analyzed with the PlantScreen$^{TM}$ Phenotyping System, Photon Systems Instruments (PSI), Czech Republic. Plants were sowed in PSI standard pots and imaged at day 25.

**Hypocotyl elongation assay**. Seeds were sown on MS plates and stratified at 4 °C for 2–3 days. Seedlings were grown vertically in long day conditions (16 h light) for 4 days then transferred to treatment plates (MS with chemicals) and grown vertically for another 3 days before seedlings were imaged using a Zeiss Stemi 2000-C stereo microscope and measured using the ImageJ software (http://rsbweb.nih.gov/ij/index.html).

**Shoot characterization**. For shoot size, inflorescence height, distance and angles between siliques, and flowering time measurements, plants were sown on soil, one plant per pot, and measurements were taken at time points indicated in figure legends with at least 12 plants for each genotype. Phyllotactic patterns were assessed on fully grown inflorescence stems of 7-week-old plants as described in Peaucelle et al.[63]. Divergence angles and internode length were measured simultaneously.

**Hormone treatment assays**. All the lines were sown on 1/2 MS plates and incubated in a Percival growth chamber for 4 days after the cold stratification (4 °C for 2 days). Seedlings were transferred to 1/2 MS plates supplemented with hormone: 1 µM trans-ABA, 1 µM CK (t-Zeatin, trans-isomer), 5 µM GA$_3$, 250 nM IAA (root assays), 5 µM IAA (hypocotyl assays), or 50 µM JA. After 3-day-treatment, plates were scanned and root length was measured using a Fiji system[64].

**Construction of phylogenetic tree and network construction**. A phylogenetic tree of ABCB family members based on protein sequence was constructed using sequences alignment by Clustal W (software http://www.clustal.org/clustal2/). One neighbor-joining phylogenetic tree was constructed using MEGA7.0 (Molecular Evolutionary Genetics Analysis) software, with 1000 of bootstrap replications. The network graph (Fig. 2) was visualized using Cytoscape (http://www.cytoscape.org/).

**Statistical analysis**. Two-tailed Student's *t*-test was performed whenever two groups were compared. Statistical significance was determined at $P < 0.01$ unless otherwise indicated. All pairs, Tukey HSD in JMP software (JMP Statistical Discovery) was used for Nanostring statistical analysis.

**Fluorescent imaging**. Seedlings were imaged on a laser scanning confocal microscope (Zeiss LSM 780) with argon laser and GaAsP detector. For protoplast imaging, protoplasts were stained in 100 µM FM4-64 for 1 min and washed by W5 solution and imaged within 80 min. For root imaging, seedlings were stained in 10 mg/L propidium iodide for 1 min, rinsed, and mounted in water.

**Scanning electron microscopy**. Scanning electron microscopy was conducted as described[65], with the following exception: samples were fixed in 2.5% glutaraldehyde in PBS. After washing, samples were dehydrated by successive ethanol treatments. Following critical point drying (Balzer's critical point drier), the samples were mounted on aluminum stubs and sputter-coated (SC7620, Quorum) with gold. Images were captured using a Jeol JCM-6000 scanning electron microscope.

**Histochemical GUS-staining**. The GUS assay was performed as previously described[66] with minor modifications. Plant tissues were incubated for approximately 16 h at 37 °C in 100 mM sodium phosphate buffer (pH 7.0) containing 0.1% Triton X-100, 1 mM 5-bromo-4-chloro-3-indolyl-β-D-glucuronic acid cyclohexylammonium salt (X-gluc, Sigma-Aldrich), 2 mM potassium ferricyanide, and 2 mM potassium ferrocyanide. Tissues were immersed with 70% ethanol until transparent. GUS-stained tissues were examined and imaged using Zeiss Stemi 2000-C stereomicroscope. Images were captured using ZEN software (Zeiss).

**Auxin transport assays**. Polar auxin transport assays in *Arabidopsis* inflorescence stems were measured using [$^{14}$C]indole acetic acid ([$^{14}$C]IAA) using a modification of a procedure described in Okada et al.[67]. When *Arabidopsis* inflorescence stems were 10–15 cm long, 2.5-cm long segments from the apical tip (not including floral parts) were harvested. The apical/basal side of each segment was then submerged in 30 µl of 0.25 MS solution (0.5% sucrose, pH 5.7) containing 1.8 µM [$^{14}$C]IAA and 250 nM IAA and incubated at room temperature for 4 h. Based on the orientation of the inflorescence segment within the tube, acropetal (polar auxin transport) or basipetal (control) auxin transport was measured. After incubation, the segment was removed and the last 5 mm of the non-submerged end was excised and placed into 3 ml scintillation liquid (Opti-Fluor). The samples were allowed to sit for at least 24 h in darkness before being counted in a liquid scintillation counter. The amount of [$^{14}$C]IAA transported to the end of the segment was reported as CPM. Radiolabeled [$^3$H]IAA; (ARC 1112), [$^{14}$C]BA; (ARC 0186A) and [$^{14}$C]malic acid; (ARC 0771) export from *Arabidopsis* protoplasts was analyzed as described previously[68]. Relative export of radioactivity from protoplasts was calculated as follows:

$$\left( \frac{(\text{radioactivity in the protoplasts at time } t = x \min) - - (\text{radioactivity in the protoplasts at time } t = 0)}{(\text{radioactivity in the protoplasts at } t = 0 \min)} \right) \times (100\%)$$

Polar auxin transport assays based on *pDR5:GUS* expression were performed in inflorescence stems of *Arabidopsis* plants expressing *amiR1334* in the *pDR5:GUS* background. The assay was performed on the F$_1$ generation of the *amiR1334* × *pDR5:GUS* cross and WT crossed with *pDR5:GUS* plants as a control. When *Arabidopsis* inflorescence stems were 10–15 cm long, 2.5 cm long segments from the apical tip (not including floral parts) were harvested. The apical side of each segment was then submerged in 30 µl of 0.25 MS solution (0.5% sucrose, pH 5.7) containing 5 µM IAA and incubated at room temperature for the duration of 5.5 h. After incubation, an overnight GUS staining was carried.

**Nanostring and quantitative RT-PCR.** Total RNA was isolated from the indicated plant materials using RNeasy Plant Mini Kit (QIAGN 74,904). For NanoString, 66 NanoString probes were synthesized by NanoString Technologies Inc., Seattle, USA[69]. The custom-designed probes included a 100-bp region targeting the mRNA, with sequence-specific, fluorescent-barcoded probes for each target. Probes and 100 ng total RNA were hybridized overnight at 65 °C according to the manufacturer's protocol[69]. A NanoString nCounter Digital Analyzer (NanoString Technologies, Seattle, USA) was used to count the digital barcodes representing the number of transcripts. The resulting data were normalized again with the geometric mean of the housekeeping genes (PP2A, CBP20, and ACT2). Nanostring probes are listed in Supplementary Table 6.

## Data availability

The authors declare that the data supporting the findings of this study are available within the paper and its Supplementary Information files. Generated plasmids and *Arabidopsis* lines are available from the corresponding author upon request.

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

## Acknowledgements

We thank Dr. Guilia Meshulam (TAU) for assistance in plant phenotyping and Laurence Charrier for transport assay assistance. This work was supported by grants from the Israel Science Foundation (1832/14 to E.S.) and (2158/14 to E.S.), the German Israeli Foundation for Scientific Research and Development (I-236-203.17-2014 to E.S.), the Human Frontier Science Program (HFSP—RGY0075/2015 to E.S. and H.H.N.-E.), by European Research Council Starting Grant (757683-RobustHormoneTrans to E.S.), the PBC postdoc fellowship (to Y.Z.), the National Science Foundation (MCB-1616236; J.I. S.), the National Institutes of Health (GM060396-ES010337; J.I.S.), the Swiss National Funds (grant 31003A_165877to M.G.) and the I-CORE Program of the Planning and Budgeting Committee and The Israel Science Foundation (grant no. 757/12; H.F.).

## Author contributions

Y.Z. and E.S. conceived and designed the study and wrote the manuscript. Y.Z. performed the research. V.N. carried out parts of the experiments presented in Figs. 3 and 4. O.P. carried out the amiRNA library transformation and the Nanostring experiments. M. O. assisted in ABCB6 cloning and imaging. N.W. performed hormone transport assays. I. T. assisted with library transformation and data analysis. M.D.D. and P.H. carried out auxin transport assays and the ABCB6 subcellular localization. O.R. performed the SEM experiments. F.H., H.F., J.I.S., H.H.N.-E., and M.G. designed and supervised the work and edited the manuscript. F.H. and J.I.S. also provided amiRNA library information prior to publication. All authors discussed the results and commented on the manuscript.

## Additional information

**Competing interests:** The authors declare no competing interests.

