## [Peer Review File · Nature Communications]

Reviewers' comments:

Reviewer #1 (Remarks to the Author):

The manuscript from Zhang et al presents an interesting approach to identifying new transporters that are associated with developmental processes. The authors used an amiRNA library and a screen for variations in photosynthesis, plant color, plant morphology and hyper-/in-sensitivity to hormone treatment to identify plant hormone transporters. From this screen the authors identified amiR1334 that displays reduced mature plant height and altered leaf morphology phenotypes that might be associated with altered auxin response and/or distribution. The authors believe the amiRNA line targets primarily gene products encoding the ATP binding cassette ABCB6 and ABCB20. Plant ABCB transporters have been reported to mobilize simple and aromatic organic acids with varying specificities. ABCB1,4,19, and 21 have been characterized as auxin transporters. ABCB11/12 and 14 have been implicated as auxin transporters, but ABCB14 is also a well characterized malate transporter. Reports from one of these authors and others suggest that multiple factors may regulate the type of substrates mobilized by ABCBs. The authors also duplicated the many previous reports indicating phylogenetic relatedness and possible structural similarity of ABCB6 and 20 to ABCB1 and 19, although the structural basis of substrate specificity in ABCB proteins remains theoretical at best. So, the rationale behind the report is reasonable, despite reports that ABC20 is a pseudogene and an absence of ABCB20 in EST and cDNA collections. Further, the Affymetrix positions attributed to ABCB20 appears to contain four elements that are identical to other ABCB sequences including ABCB6, so expression reported with this method may be questioned.

While single *abcb6* or *abcb20* T-DNA insertion lines did not display significant growth phenotypes, double *abcb6abcb20* mutants displayed mature shoot and leaf phenotypes reminiscent of amiR1334 line. It would be important to know if these differences reflect a delay in growth or a difference in final height and/ or mass. Additionally, the authors report alterations in silique phyllotaxy and some silique twisting. Considering the authors own account that they cannot detect ABCB20 expression, it would help to have a second set of alleles for the double mutants and/or verification that ABCB1 and 19 expression are not impacted in the double mutant.

The *abcb6* (GABI_401D12) has an insertion after NT 1381 (exon), end of TMH6 in protein. The authors do not appear to test whether there is a partial mRNA. In at least one other cases (*abcb19-5*) partial products appear to result in co-suppression of other ABCB transporters including ABCB6.

It appears that a second *abcb20* allele could be used as two GABI_KAT accessions are listed in TAIR
GABI_520G10.020047: intron 8, in linker domain between NBD1 and TMH7 of protein
GABI_520G10.020203: insertion after NT 1947 (exon), end of NBD1 in protein
Specific GABI_KAT line used not specified in text but according to Fig 3D it is GABI_520G10.020203.

No major differences in root morphology were reported for amiRNA, single, or double mutants. Inferred auxin conductance in inflorescence stems in the amiRNA line was shown to be reduced using the DR5:GUS reporter. Better resolution images would be required to determine if these transport assays make sense as it is not clear that the DR5 signal is associated with vascular strands where auxin transport should occur. These could be correlated with ABCB6Pro:GUS expression in the inflorescence stems (also not shown). Perhaps QRT-PCR of ABCB6 and ABCB20 expression in the segments tested is in order as well. Also, 3H-IAA transport in *abcb6abcb20* is reduced ~50% in inflorescence stems. There is also concern about the reported expression of NLS-YFP under the control of ABCB6 promoter, as the image shown appears to have signals in root epidermal cells that is non-nuclear. This requires some attention

Leaf mesophyll protoplasts from *abcb6* and *abcb20* showed 54% and 70% reductions in auxin transport, while the *abcb6abcb20* and *amiRNA* was reduced by 73% and 66%. Again, this is interesting evidence, but the expression of ABCB6 and ABCB20 should be confirmed in the wild type protoplasts. Further, the transport assays don't show the kinetics of the transport assays usually shown by the authors. Benzoic acid controls are useful, but competition assays with malate and/or cinnamic acid might be in order.

It is unfortunate that the authors were unable to clone coding or genomic sequences from ABCB20, but is consistent with reports that ABCB20 expression is undetectable.

As would be expected, plasma membrane localization of p35S:ABCB6-YFP (CDS) in *Arabidopsis* protoplasts and infiltrated tobacco leaves is shown. The evidence is not particularly convincing, but the localization to the PM is highly likely.

pABCB6:GUS-GFP showed strong signals in the vasculature of true leaves, and the differentiated root vasculature tissues. A weaker GUS signal was observed in the cotyledons and hypocotyl. Expression was insensitive to IAA treatment by RT-PCR and GUS-GFP.

A major concern is that the authors have not shown any results regarding ABCB1 and 19 expression in the mutant and *ami* lines. This is particularly important, as ABCB19 is highly expressed in response to auxin treatment, and the phenotype of *abcb1* mutants are largely masked due to compensatory expression of ABCB19. There is also concern that the observed effects may be a result of knockdown of ABCB19. The authors did not check for knockdown of ABCB6 and ABCB20 in *amiR1334*, expression in respective mutant backgrounds, or compensation/knockdown of other ABCBs. The reduction in height at 100 μ E is equivalent to what is observed in the *abcb19/ mdr1-101*. Further, in Fig. 3F the *amiR1334* height is shown as 75-80% of WT but in Fig. S6A it looks to be < 10%. Could this be variability due to different levels of expression of other ABCBs (particularly ABCB1 and ABCB19)? The upright silique angles are also reminiscent of *pgp19-5* and *mdr1-101*.

It would also help to know if the silique phyllotaxy is measured from the point of emergence or the angle of silique itself relative to neighboring siliques? From Fig 3H, the point of pedicel emergence from the inflorescence does not appear to be different.

Additional notes on technical basis of concerns about *amiR1334*:

amiR1334: ACGCCGACTGCTGTGATCAG

B6 and B20 83.7% mRNA similarity

amiR1334 targets NT 487-506 in ABCB6 mRNA, 493-512 in ABCB20 mRNA

According to WMD3:

2 mismatches: ABCB6 and ABCB20

5 mismatches: ABCB1

6 mismatches: ABCB18, ABCB19, ABCB2, ABCB5

Targets according to WMD3:

amiR1334: ACGCCGACTGCTGTGATCAG

2 mismatches

ABCB6: Possible candidate. Low binding affinity: Only 30.92% of optimal dG

Hybridization energy: -14.79kcal/mol (30.92 %)

Target gene 5'->3'/699-718 AGGCAGACTGCTGTGATCAG

amiRNA (rev. complement) ACGCCGACTGCTGTGATCAG

ABCB20: Possible candidate. Low binding affinity: Only 30.92% of optimal dG

Hybridization energy: -14.79kcal/mol (30.92 %)

Target gene 5'->3'/493-512 AGGCAGACTGCTGTGATCAG

amiRNA (rev. complement) ACGCCGACTGCTGTGATCAG

5 mismatches

ABCB1: Low binding affinity: Only 33.72% of optimal dG

Hybridization energy: -16.13kcal/mol (33.72 %)

Target gene 5'->3'/743-762 CCGTTGATTGCTGTGATCGG

amiRNA (rev. complement) ACGCCGACTGCTGTGATCAG

6 mismatches

ABCB18:

Hybridization energy: -15.88kcal/mol (33.19 %)

Target gene 5'->3'/2971-2990 ACACGGCCTGATGTGATTAT

amiRNA (rev. complement) ACGCCGACTGCTGTGATCAG

ABCB19:

Hybridization energy: -14.37kcal/mol (30.04 %)

Target gene 5'->3'/3652-3671 AAGAGGAGTGCAGTTATCAG

amiRNA (rev. complement) ACGCCGACTGCTGTGATCAG

ABCB2:

Hybridization energy: -13.95kcal/mol (29.16 %)

Target gene 5'->3'/2740-2759 TCGCTGCCTTCTGTGCTGAG

amiRNA (rev. complement) ACGCCGACTGCTGTGATCAG

ABCB5:

Hybridization energy: -11.46kcal/mol (23.95 %)

Target gene 5'->3'/1177-1196 GGGAAAGTCTACTGTGATCAG

amiRNA (rev. complement) ACGCCGACTGCTGTGATCAG

7 mismatches

ABCB3:

Hybridization energy: -12.84kcal/mol (26.84 %)

Target gene 5'->3'/1162-1181 GGGAAATCTTCTGTGATCAG

amiRNA (rev. complement) ACGCCGACTGCTGTGATCAG

Hybridization energy: -7.94kcal/mol (16.60 %)

Target gene 5'->3'/1603-1622 AGCCGGACTACTGTGATTGT

amiRNA (rev. complement) ACGCCGACTGCTGTGATCAG

ABCB1:

Hybridization energy: -14.85kcal/mol (31.04 %)

Target gene 5'->3'/2486-2505 ATGCTGAGCGCTGTACTCAA

amiRNA (rev. complement) ACGCCGACTGCTGTGATCAG

Hybridization energy: -9.44kcal/mol (19.73 %)
Target gene 5'->3'/252-271 TCGCCGACGGATTAGATTAT
amiRNA (rev. complement) ACGCCGACTGCTGTGATCAG

ABCB20:

Hybridization energy: -10.97kcal/mol (22.93 %)
Target gene 5'->3'/1124-1143 ACTTCTACTCTTTTGATCAA
amiRNA (rev. complement) ACGCCGACTGCTGTGATCAG

In summary, this report is compelling, but requires attention to some details to strongly implicate ABCB6 and ABCB20 in developmentally-important auxin transport. The bar is higher in light of the lack of detectable expression of ABCB20. The authors might also want to make it clear that transport of auxin may not be the only function of the transporters characterized. This should not diminish the importance of the work, but could help promote a better understanding of transporter function.

Reviewer #2 (Remarks to the Author):

Zhang et al. performed a high-throughput amiRNA screening to find novel auxin-transporting ABC transporters. Among those screened, two ABCBs (6 and 20) were identified to be auxin-exporting transporters. T-DNA mutant lines were used to confirm the results from the amiRNA lines. The mutants for ABCB6 and 20 showed auxin-defective morphological phenotypes and defects in auxin transport of inflorescence stems and leaf mesophyll protoplasts. The auxin transport assay with protoplasts revealed that those ABCBs have an auxin-exporting activity at the plasma membrane.

I believe that the amiRNA screening process should be a lot of work and was demonstrated to be a promising tool to find the function of multigene family members. The experiments to confirm the biological role of those ABCBs were carefully performed and reasonably presented.

Although several ABCB transporters have been already reported to function as auxin transporters, this study identified two more novel ABCB auxin transporters and showed that cellular auxin movement is achieved by a lot more transporting proteins, which could reflect the complexity in auxin-mediated morphogenesis of plants.

Discovery of the novel auxin transporters and demonstration of the usefulness of amiRNA screening should interest the readers of Nature Communications.

Because this study focuses on the auxin-transporting ABCBs, several early key works, where auxin-transporting ABCBs were characterized but were missed here, need to be appreciated in the citation.

Reviewers' comments:

Reviewer #1 (Remarks to the Author):

The manuscript from Zhang et al presents an interesting approach to identifying new transporters that are associated with developmental processes. The authors used an amiRNA library and a screen for variations in photosynthesis, plant color, plant morphology and hyper-/in-sensitivity to hormone treatment to identify plant hormone transporters. From this screen the authors identified amiR1334 that displays reduced mature plant height and altered leaf morphology phenotypes that might be associated with altered auxin response and/or distribution. The authors believe the amiRNA line targets primarily gene products encoding the ATP binding cassette ABCB6 and ABCB20. Plant ABCB transporters have been reported to mobilize simple and aromatic organic acids with varying specificities. ABCB1,4,19, and 21 have been characterized as auxin transporters. ABCB11/12 and 14 have been implicated as auxin transporters, but ABCB14 is also a well characterized malate transporter. Reports from one of these authors and others suggest that multiple factors may regulate the type of substrates mobilized by ABCBs. The authors also duplicated the many previous reports indicating phylogenetic relatedness and possible structural similarity of ABCB6 and 20 to ABCB1 and 19, although the structural basis of substrate specificity in ABCB proteins remains theoretical at best. So, the rationale behind the report is reasonable, despite reports that ABC20 is a pseudogene and an absence of ABCB20 in EST and cDNA collections. Further, the Affymetrix positions attributed to ABCB20 appears to contain four elements that are identical to other ABCB sequences including ABCB6, so expression reported with this method may be questioned.

While single *abcb6* or *abcb20* T-DNA insertion lines did not display significant growth phenotypes, double *abcb6abcb20* mutants displayed mature shoot and leaf phenotypes reminiscent of amiR1334 line. It would be important to know if these differences reflect a delay in growth or a difference in final height and/ or mass.

We have documented *abcb6-1 abcb20-1*, *abcb6-1 abcb20-2* and *amiRNA-1334* over time and show that all lines are significantly smaller/shorter, with delayed growth over time and in their final stage (Fig. 3 Fig. S7, S8, S9).

Additionally, the authors report alterations in silique phyllotaxy and some silique twisting. Considering the authors own account that they cannot detect ABCB20 expression, it would help to have a second set of alleles for the double mutants and/or verification that ABCB1 and 19 expression are not impacted in the double mutant.

We obtained and genotyped an additional, independent *abcb20* allele (GABI_387F09) termed *abcb20-2*. The *abcb6-1 abcb20-2* double mutant showed identical phenotypes to *abcb6-1 abcb20-1* double mutant and *amiRNA1334* line. *ABCB6* and *ABCB20* are significantly down

regulated in both alleles of *abcb6 abcb20* double mutant and *amiRNA1334* line. *ABCB1* and *ABCB19* expression did not change in these lines (Fig. S11).

The *abcb6* (GABI_401D12) has an insertion after NT 1381 (exon), end of TMH6 in protein. The authors do not appear to test whether there is a partial mRNA. In at least one other cases (*abcb19-5*) partial products appear to result in co-suppression of other ABCB transporters including ABCB6.

We tested the expression levels of all ABCBs in the background of *abcb6-1 abcb20-1* double mutant, *abcb6-1 abcb20-2* double mutant and *amiRNA1334* line. The results show that *ABCB6* and *ABCB20* are the only two genes which were downregulated in all three genotypes (*abcb6 abcb20* loss of function lines). (Fig. S10, S11).

It appears that a second *abcb20* allele could be used as two GABI_KAT accessions are listed in TAIR
GABI_520G10.020047: intron 8, in linker domain between NBD1 and TMH7 of protein
GABI_520G10.020203: insertion after NT 1947 (exon), end of NBD1 in protein
Specific GABI_KAT line used not specified in text but according to Fig 3D it is GABI_520G10.020203.

We obtained and genotyped the requested independent *abcb20-2* allele (GABI_387F09) termed *abcb20-2*. While the single *abcb20-2* mutant did not show a significant phenotype (Fig.S9), the *abcb6-1 abcb20-2* double mutant showed identical phenotypes to *abcb6-1 abcb20-1* double mutant and *amiRNA1334* line. The details of each T-DNA lines are now described in text (results and methods).

No major differences in root morphology were reported for *amiRNA*, single, or double mutants. Inferred auxin conductance in inflorescence stems in the *amiRNA* line was shown to be reduced using the DR5:GUS reporter. Better resolution images would be required to determine if these transport assays make sense as it is not clear that the DR5 signal is associated with vascular strands where auxin transport should occur. These could be correlated with ABCB6Pro:GUS expression in the inflorescence stems (also not shown).

We now show *pDR5:GUS* stem sections (Fig. S13A).

Perhaps QRT-PCR of ABCB6 and ABCB20 expression in the segments tested is in order as well. Also, 3H-IAA transport in *abcb6abcb20* is reduced ~50% in inflorescence stems.

We tested *ABCB6* and *ABCB20* expression in the segmented stems at identical conditions as the transport assays. *ABCB6* and *ABCB20* are significantly down regulated in the *abcb6 abcb20* double mutant compared to WT (Fig. S13).

There is also concern about the reported expression of NLS-YFP under the control of ABCB6 promoter, as the image shown appears to have signals in root epidermal cells that is non-nuclear. This requires some attention

We added new *pABC6:NLS-YFP* root images with higher quality (Fig. 4).

Leaf mesophyll protoplasts from *abcb6* and *abcb20* showed 54% and 70% reductions in auxin transport, while the *abcb6abcb20* and *amiRNA* was reduced by 73% and 66%. Again, this is interesting evidence, but the expression of ABCB6 and ABCB20 should be confirmed in the

wild type protoplasts. Further, the transport assays don't show the kinetics of the transport assays usually shown by the authors. Benzoic acid controls are useful, but competition assays with malate and/or cinnamic acid might be in order.

Nanostring analyses confirmed that all relevant ABCB genes are expressed in shoot WT tissue classically used for the generation of leaf protoplasts. Further, we carried out malic acid transport assays. The results showed no significant change in malic acid transport when comparing WT to *abcb6-1 abcb20-1* double mutant and *amiRNA1334* (Fig. S14).

It is unfortunate that the authors were unable to clone coding or genomic sequences from ABCB20, but is consistent with reports that ABCB20 expression is undetectable.

We were able to clone *pABCB20* and now report on its expression pattern using the GUS reporter (Fig. 4).

As would be expected, plasma membrane localization of p35S:ABCB6-YFP (CDS) in Arabidopsis protoplasts and infiltrated tobacco leaves is shown. The evidence is not particularly convincing, but the localization to the PM is highly likely.

These are high quality images obtained independently by two labs (Geisler and Shani labs). The stable transformation *p35S:ABCB6-YFP* plants we generated were highly variant and silenced.

pABCB6:GUS-GFP showed strong signals in the vasculature of true leaves, and the differentiated root vasculature tissues. A weaker GUS signal was observed in the cotyledons and hypocotyl. Expression was insensitive to IAA treatment by RT-PCR and GUS-GFP.

A major concern is that the authors have not shown any results regarding ABCB1 and 19 expression in the mutant and ami lines. This is particularly important, as ABCB19 is highly expressed in response to auxin treatment, and the phenotype of *abcb1* mutants are largely masked due to compensatory expression of ABCB19. There is also concern that the observed effects may be a result of knockdown of ABCB19. The authors did not check for knockdown of ABCB6 and ABCB20 in *amiR1334*, expression in respective mutant backgrounds, or compensation/knockdown of other ABCBs. The reduction in height at 100 uE is equivalent to what is observed in the *abcb19/ mdr1-101*. Further, in Fig. 3F the *amiR1334* height is shown as 75-80% of WT but in Fig. S6A it looks to be < 10%. Could this be variability due to different levels of expression of other ABCBs (particularly ABCB1 and ABCB19)? The upright silique angles are also reminiscent of *pgp19-5* and *mdr1-101*.

We tested the expression levels of all ABCBs in the background of *abcb6-1 abcb20-1* double mutant and *amiRNA1334* line. *ABCB6* and *ABCB20* are significantly down regulated in both alleles of *abcb6 abcb20* double mutant and *amiRNA1334* line (Fig.3, Fig.S11, S14). *ABCB1* and *ABCB19* expression does not change in these lines (Fig. S10). No significant change was observed for other ABCBs in both *abcb6 abcb20* double mutant and *amiRNA1334* line (Fig. S11).

It would also help to know if the silique phyllotaxy is measured from the point of emergence or the angle of silique itself relative to neighboring siliques? From Fig 3H, the point of pedicel emergence from the inflorescence does not appear to be different.

In order to address this, we measured the meristem morphogenesis of *abcb6-1 abcb20-1* double mutant compared to WT. We could not observe extreme changes although slight changes might exist. We show that the stem of *abcb6-1 abcb20-1* double mutant is twisting in high resolution (SEM) and added images to address the modification of organ initiation over time (Fig. S12).

Additional notes on technical basis of concerns about amiR1334:

amiR1334: ACGCCGACTGCTGTGATCAG

B6 and B20 83.7% mRNA similarity

amiR1334 targets NT 487-506 in ABCB6 mRNA, 493-512 in ABCB20 mRNA

According to WMD3:

2 mismatches: ABCB6 and ABCB20

5 mismatches: ABCB1

6 mismatches: ABCB18, ABCB19, ABCB2, ABCB5

Targets according to WMD3:

amiR1334: ACGCCGACTGCTGTGATCAG

2 mismatches

ABCB6: Possible candidate. Low binding affinity: Only 30.92% of optimal dG

Hybridization energy: -14.79kcal/mol (30.92 %)

Target gene 5'->3'/699-718 AGGCAGACTGCTGTGATCAG

amiRNA (rev. complement) ACGCCGACTGCTGTGATCAG

ABCB20: Possible candidate. Low binding affinity: Only 30.92% of optimal dG

Hybridization energy: -14.79kcal/mol (30.92 %)

Target gene 5'->3'/493-512 AGGCAGACTGCTGTGATCAG

amiRNA (rev. complement) ACGCCGACTGCTGTGATCAG

5 mismatches

ABCB1: Low binding affinity: Only 33.72% of optimal dG

Hybridization energy: -16.13kcal/mol (33.72 %)

Target gene 5'->3'/743-762 CCGTTGATTGCTGTGATCGG

amiRNA (rev. complement) ACGCCGACTGCTGTGATCAG

6 mismatches

ABCB18:

Hybridization energy: -15.88kcal/mol (33.19 %)

Target gene 5'->3'/2971-2990 ACACGGCCTGATGTGATTAT

amiRNA (rev. complement) ACGCCGACTGCTGTGATCAG

ABCB19:

Hybridization energy: -14.37kcal/mol (30.04 %)

Target gene 5'->3'/3652-3671 AAGAGGAGTGCAGTTATCAG

amiRNA (rev. complement) ACGCCGACTGCTGTGATCAG

ABCB2:

Hybridization energy: -13.95kcal/mol (29.16 %)

Target gene 5'->3'/2740-2759 TCGCTGCCTTCTGTGCTGAG

amiRNA (rev. complement) ACGCCGACTGCTGTGATCAG

ABCB5:

Hybridization energy: -11.46kcal/mol (23.95 %)

Target gene 5'->3'/1177-1196 GGGAAAGTCTACTGTGATCAG

amiRNA (rev. complement) ACGCCGACTGCTGTGATCAG

7 mismatches

ABCB3:

Hybridization energy: -12.84kcal/mol (26.84 %)

Target gene 5'->3'/1162-1181 GGGAAATCTTCTGTGATCAG

amiRNA (rev. complement) ACGCCGACTGCTGTGATCAG

Hybridization energy: -7.94kcal/mol (16.60 %)

Target gene 5'->3'/1603-1622 AGCCGGACTACTGTGATTGT

amiRNA (rev. complement) ACGCCGACTGCTGTGATCAG

ABCB1:

Hybridization energy: -14.85kcal/mol (31.04 %)

Target gene 5'->3'/2486-2505 ATGCTGAGCGCTGTACTCAA

amiRNA (rev. complement) ACGCCGACTGCTGTGATCAG

Hybridization energy: -9.44kcal/mol (19.73 %)

Target gene 5'->3'/252-271 TCGCCGACGGATTAGATTAT

amiRNA (rev. complement) ACGCCGACTGCTGTGATCAG

ABCB20:

Hybridization energy: -10.97kcal/mol (22.93 %)

Target gene 5'->3'/1124-1143 ACTTCTACTCTTTTGGATCAA

amiRNA (rev. complement) ACGCCGACTGCTGTGATCAG

In summary, this report is compelling, but requires attention to some details to strongly implicate ABCB6 and ABCB20 in developmentally-important auxin transport. The bar is higher in light of the lack of detectable expression of ABCB20. The authors might also want to make it clear that transport of auxin may not be the only function of the transporters characterized. This should not diminish the importance of the work, but could help promote a better understanding of transporter function.

We added several important results to the current version, including the detection of *ABCB20* expression by Nanostring and GUS, in addition to the new *abcb20-2* allele. We modified the text to clarify that auxin may not be the only substrate of these new ABCBs but excluded that they transport malic acid, like described for ABCB14 (see discussion).

Reviewer #2 (Remarks to the Author):

Zhang et al. performed a high-throughput amiRNA screening to find novel auxin-transporting ABC transporters. Among those screened, two ABCBs (6 and 20) were identified to be auxin-exporting transporters. T-DNA mutant lines were used to confirm the results from the amiRNA lines. The mutants for ABCB6 and 20 showed auxin-defective morphological phenotypes and defects in auxin transport of inflorescence stems and leaf mesophyll protoplasts. The auxin transport assay with protoplasts revealed that those ABCBs have an auxin-exporting activity at the plasma membrane.

I believe that the amiRNA screening process should be a lot of work and was demonstrated to be a promising tool to find the function of multigene family members. The experiments to confirm the biological role of those ABCBs were carefully performed and reasonably presented.

Although several ABCB transporters have been already reported to function as auxin transporters, this study identified two more novel ABCB auxin transporters and showed that cellular auxin movement is achieved by a lot more transporting proteins, which could reflect the complexity in auxin-mediated morphogenesis of plants.

Discovery of the novel auxin transporters and demonstration of the usefulness of amiRNA screening should interest the readers of Nature Communications.

Because this study focuses on the auxin-transporting ABCBs, several early key works, where auxin-transporting ABCBs were characterized but were missed here, need to be appreciated in the citation.

We revised the text and added several important citations.

REVIEWERS' COMMENTS:

Reviewer #1 (Remarks to the Author):

The authors are thanked for taking the comments of this reviewer seriously and addressing all of the issues raised. The manuscript is greatly improved by the additional work and now makes a compelling case for the role of ABCB6 and 20 in regulating auxin transport